# Efficient High-Resolution Image Editing with Hallucination-Aware Loss and Adaptive Tiling

## Abstract

High-resolution (4K) image-to-image synthesis has become increasingly important for mobile applications. Existing diffusion models for image editing face significant challenges, in terms of memory and image quality, when deployed on resource-constrained devices. In this paper, we present *MobilePicasso*, a novel system that enables efficient image editing at high resolutions, while minimising computational cost and memory usage. *MobilePicasso* comprises three stages: (i) performing image editing at a standard resolution with hallucination-aware loss, (ii) applying latent projection to overcome going to the pixel space, and (iii) upscaling the edited image latent to a higher resolution with adaptive context-preserving tiling. Our user study with 46 participants reveals that *MobilePicasso* not only improves image quality by 18-48% but reduces hallucinations by 14-51% over existing methods. *MobilePicasso* demonstrates significantly lower latency, e.g., up to $55.8\times$ speed-up, yet with a small increase in runtime memory, e.g., a mere 9% increase over prior work. Surprisingly, the on-device runtime of *MobilePicasso* is observed to be faster than a server-based high-resolution image editing model running on an A100 GPU.

## 1 Introduction

Hallucinations are unrealistic objects or elements generated by diffusion models that were not intended by the edit instruction, such as distorted faces, floating objects, or implausible scenes. Diffusion-based image-to-image (I2I) synthesis has emerged as a powerful tool for image editing (Sheynin et al., 2023), enabling millions of users to modify images through natural language instructions (Brooks et al., 2022; Zhang et al., 2023a; Wasserman et al., 2024) such as "Rainy weather in the background" or "Make it Pixar style." However, these diffusion models (DMs) are large-scale and computationally intensive, typically requiring cloud-based solutions and do not adequately address users' privacy.

Enabling on-device image editing with DMs at native mobile resolution presents significant challenges. Firstly, the maximum resolution supported by most models remains limited. Latest models such as SDXL (Podell et al., 2023), SD3-series (Esser et al., 2024), Flux.1-series (Black Forest Labs, 2024) models support resolutions up to $1024 \times 1024$, falling short of real-world applications for phones, tablets, and TVs. Popular mobile screen resolutions range between $1080 \times 2640$ and $1440 \times 3120$, while Tablets and TVs require even higher resolutions. While recent works like DemoFusion (Du et al., 2023) can generate 4K images, its nine-minute processing time on A100 GPUs makes it completely impractical for resource-constrained devices, where such processing delays could severely impact user experience. Secondly, existing I2I generation models, such as InstructPix2Pix (IP2P) (Brooks et al., 2022), MagicBrush (Zhang et al., 2023a), and PIPE (Wasserman et al., 2024), often produce artefacts and hallucinations, i.e., distorted and implausible objects such as a floating lamp (Liang et al., 2024; Kirstain et al., 2023) (see Figure 5) even at a standard resolution ($512 \times 512$), which becomes more problematic in higher resolutions (see Figure 1 for more examples). Careful human evaluations show that around 30% of the generated images using our test set (Section 4.1) contain hallucinations. Thirdly, mobile devices have limited computational and memory resources, where diffusion model deployment can be problematic. For example, measurements on Snapdragon 8 Gen 2 NPU (on Samsung Galaxy S23) reveal that system overheads increase dramatically with input image resolution (Figure 1) and resolutions over $1024 \times 1024$ often lead to an out-of-memory (OOM) error.

While a tiling approach (von Platen et al., 2022; Song et al., 2024) can address memory constraints by processing images in smaller segments or tiles (see Figures 3a and 3b for examples of tiles), it introduces significant new challenges. Processing overlapping tiles with DMs incurs substantial computational overhead that scales quadratically with the overlap ratio between successive tiles.

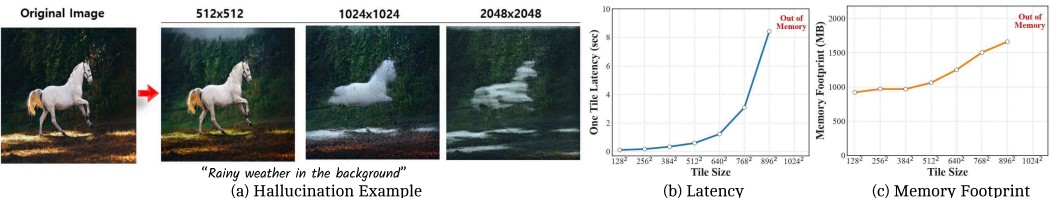

(a) Hallucination Example        (b) Latency        (c) Memory Footprint

Figure 1: Figure (a) shows the typical examples of the effect of image resolutions on I2I generation. As image resolutions get larger starting from $512 \times 512$ to $2048 \times 2048$, I2I image edit models such as IP2P are often unable to produce realistic images that align well with the edit prompt. The measurements of latency (b) and memory (c) to run U-Net on a single tile according to various tile sizes on Snapdragon 8 Gen 2 NPU.

Moreover, small overlaps compromise image quality, producing more artefacts/glitches due to a limited context from neighboring tiles (see Figure 6).

To address these challenges, we present *MobilePicasso*, which enables the high-resolution (4K) on-device I2I generation, addressing computational and memory constraints.

We introduce a **3-stage hybrid pipeline** that *formulates the challenging high-resolution image editing task into three sub-tasks that are easier to tackle*: (1) image editing at standard resolutions ($512 \times 512$), (2) learnable latent projection, and (3) upscaling stages. Compared to direct image editing at high resolutions, image editing at a standard resolution natively addresses hallucination within the memory constraint of mobile devices as demonstrated in Figures 1 and 6. Also, our learnable latent projection and upscaling enable most operations happening in latent space, further reducing computation by avoiding encoding and decoding steps that commonly exist in image editing (Brooks et al., 2022) and image upscaling (Noroozi et al., 2024) (Section 3.1).

*MobilePicasso* establishes the first comprehensive hallucination aware training framework for mobile image editing by combining **Hallucination-Aware Loss and Data Filtering** that achieves 14-51% hallucination reduction compared to existing methods (IP2P, MagicBrush, PIPE) (Section 3.2). Through an extensive user study with 46 participants, we demonstrate that *MobilePicasso* significantly outperforms all baselines: *MobilePicasso* achieves 18-48% and 22-54% improvement in overall image quality and text alignment, respectively, compared to existing methods (see Table 2).

We develop **Adaptive Context-Preserving Tiling (ACPT)** to address computational overheads and glitches in the upscaling stage by leveraging our proposed *Adjacent Padding* that provides consistent context around tiles without overlaps (Section 3.3). Additionally, we implement a **model/system co-design approach** to further optimise on-device resource usage, enhancing latency and memory efficiency by carefully analysing tile size and overlap ratios (Section 3.4).

We implemented *MobilePicasso* on a mobile device (Samsung Galaxy S23 with a Hexagon NPU). Our extensive experiments demonstrate that *MobilePicasso* achieves (1) superior high-resolution image quality over prior works that require running the generative model on powerful GPUs, both quantitatively (see Table 5) and qualitatively (see Figure 6), (2) rapid image editing, taking only 42 seconds on Galaxy S23, achieving up to $55.8\times$ speed-up compared to baselines using tiling with overlaps. Surprisingly, *MobilePicasso* is even $4.71\times$ faster than server-based baseline running on A100 GPU, and (3) affordable memory usage of 1.15 GB, significantly smaller ($71.9\times$) than server-based image editing (see Table 3), throughout the execution of the image editing and upscaling stages with our efficient on-device implementation (Section 4). *Our work paves the way for practical real-world image editing applications on mobile devices by drastically minimising resource overheads without compromising image editing quality.*

## 2 RELATED WORK

**Diffusion-based Image Editing.** Image editing tasks have long been investigated in computer vision and graphics communities (Oh et al., 2001; Pérez et al., 2023). The emergence of text-to-image DMs (Song et al., 2021; Rombach et al., 2022; Podell et al., 2023; Stability AI, 2023b; Lin et al., 2024; Lipman et al., 2022; Karras et al., 2022; Peebles & Xie, 2022; Lu et al., 2024; Esser et al., 2024) facilitated the rapid development of text-based image editing, allowing users to seamlessly edit their images with natural language. For instance, to provide a more intuitive and user-friendly tool, IP2P pioneered instruction-based image editing by constructing a large synthetic dataset consisting

of paired images generated by Prompt-to-Prompt (Hertz et al., 2022) and corresponding instruction prompts produced by GPT-3 (Brown et al., 2020) using image captions. To improve the quality of the IP2P synthetic dataset, MagicBrush (Zhang et al., 2023a) collected a manually annotated instruction-guided image editing dataset by using an online image editing tool. In addition, PIPE (Wasserman et al., 2024) introduced a large-scale realistic image-editing dataset by leveraging the insight that removing objects is simpler than adding them. This makes PIPE maintain consistency between the source (object-removed) image and the target (original/real) image. Although the edit type of PIPE is limited to only addition, its image quality is superior to all the prior works. Despite the progress in methods and datasets, state-of-the-art (SOTA) image editing methods often generate hallucinated images and struggle to closely follow edit instructions, as demonstrated in Figure 5. In this work, we improve the overall image quality of DMs by proposing hallucination-aware loss and filtering out images with artefacts from a training dataset.

**On-device Deployment of Diffusion Models.** To deploy large-scale DMs on-device, prior works examined various techniques such as low-precision quantization (e.g., 8-bit weights and activations) to reduce on-device requirements (Wang et al., 2024). SnapFusion (Li et al., 2023) decreased the number of iterations per sample generation and considered smaller and more efficient architectures. Furthermore, there is a resurgence of GANs in the context of distillation of pre-trained DMs, which allow 1-step generation, e.g., SD-Turbo (Stability AI, 2023a) (based on ADD (Sauer et al., 2023)) and MobileDiffusion (Zhao et al., 2023) (based on UFOGen (Xu et al., 2023)). However, none of the prior work focuses on high-resolution (4K) due to the strict memory constraints on mobile devices. At the same time, the image editing task has not been thoroughly investigated.

## 3 METHOD

**Formulation of Diffusion-based Image Editing.** We briefly review diffusion-based image editing (Brooks et al., 2022) and introduce our hallucination-aware loss. DMs are formulated as:

$$\mathbb{E}_{\epsilon,t}\left[\|\epsilon - f_\theta\left(\mathbf{z}_t, t\right)\|\right],$$
$$\boldsymbol{z} \sim p_{\text{data}}(\cdot),\ \boldsymbol{\epsilon} \sim \mathcal{N}(\cdot|\mathbf{0}, \boldsymbol{I}),\ t \in [0,1],\ \alpha_t > 0,\ \sigma_t > 0,$$
$$\boldsymbol{z}_t := \alpha_t \boldsymbol{z} + \sigma_t \boldsymbol{\epsilon}, \tag{1}$$

where noise $\epsilon$ increases over timesteps $t \mapsto \alpha_t^2/\sigma_t^2$ is strictly decreasing and $\alpha_1^2/\sigma_1^2 = 0$. $z$ is the encoded latent of an image $\boldsymbol{z} = \mathcal{E}(\boldsymbol{x})$ where $\mathcal{E}$ is the VAE encoder.

In I2I editing task based on DMs, $\boldsymbol{f}$, U-Net processing usually depends on two additional inputs, namely $\boldsymbol{c}_\text{T}$ and $\boldsymbol{c}_\text{I}$, encoding text and image, respectively. In particular, inspired by IP2P (Brooks et al., 2022) for image editing use case, instead of simply adding these arguments to $\boldsymbol{f}$, we employ classifier-free guidance (CFG) (Ho & Salimans, 2022), which replaces $\boldsymbol{f}$ with $\boldsymbol{g}$. Then, CFG leverages three function evaluations to construct a new approximation via:

$$\begin{aligned} \boldsymbol{g_\theta}(\boldsymbol{z}_t, t, \boldsymbol{c}_\text{I}, \boldsymbol{c}_\text{T}) :=& \boldsymbol{f_\theta}(\boldsymbol{z}_t, t, \emptyset, \emptyset) \\ &+ s_\text{I}(\boldsymbol{f_\theta}(\boldsymbol{z}_t, t, \boldsymbol{c}_\text{I}, \emptyset) - \boldsymbol{f_\theta}(\boldsymbol{z}_t, t, \emptyset, \emptyset)) \\ &+ s_\text{T}(\boldsymbol{f_\theta}(\boldsymbol{z}_t, t, \boldsymbol{c}_\text{I}, \boldsymbol{c}_\text{T}) - \boldsymbol{f_\theta}(\boldsymbol{z}_t, t, \boldsymbol{c}_\text{I}, \emptyset)) \end{aligned} \tag{2}$$

where $s_\text{I} > 0$ and $s_\text{T} > 0$ are hyper-parameters to control image aesthetics. While increasing $s_\text{I}$ encourages edited images to closely resemble input images, increasing $s_\text{T}$ facilitates edited images to follow the edit prompts closely. Through this process, U-Net takes an encoded latent as input and produces a processed latent, containing an edited image feature. The processed latent is then passed to the VAE decoder $\mathcal{D}$ and converted from latent to pixel space.

### 3.1 3-STAGE HYBRID PIPELINE

High-resolution image editing faces significant challenges, including image quality degradation at larger resolutions and resource constraints on mobile devices. To address these challenges, we propose a novel *3-stage hybrid pipeline* that breaks down the complex high-resolution image editing task into three stages, as illustrated in Figure 2: (1) standard-resolution image editing ($512 \times 512$), (2) learnable latent projection, and (3) upscaling to high resolutions (4K).

**Image Edit Stage.** *MobilePicasso* conducts image editing at a standard image resolution (e.g., $512 \times 512$) using an input image and an edit prompt. Since I2I DMs operate at an image resolution

Figure 2: The overview of *MobilePicasso*'s 3-stage hybrid pipeline, which partitions the task of high-resolution image editing into three stages: (1) image editing at standard resolution ($512^2$), (2) learnable latent projection in latent space, and (3) upscaling to higher resolutions (4K). This modular approach allows *MobilePicasso* to solve each stage effectively and efficiently for deployment.

used during pretraining the model (Brooks et al., 2022; Wasserman et al., 2024), they natively produce much fewer hallucinations as demonstrated in Figures 1 and 6, without incurring OOM errors, compared to high-resolution image editing directly. With our novel *hallucination-aware loss* used during this stage, hallucinations are reduced further.

**Learnable Latent Projection Stage.** Given a processed latent from the image editing stage, we first need to upscale it to a higher resolution space that serves as an input for the subsequent stage. This approach significantly optimises computations by performing most operations in the latent space, eliminating the need of costly encoding and decoding steps as in (Brooks et al., 2022; Noroozi et al., 2024). However, simple linear upsampling (e.g., bilinear, bicubic, and nearest) of the processed latent proves ineffective, often generating low-quality images. To address this challenge, we propose to learn projection between processed latents from the image edit stage and the upscaled encoded latents used as input during the upscaling stage. Inspired by prior work (Fu et al., 2024), we propose a lightweight learnable projection model that takes the processed latents from the image editing stage and upscales it to a higher resolution space. This projection model includes a Tiny AutoEncoder (Ollin Boer Bohan, 2024) with only 1.2M parameters and convolution layers with 6K parameters. Overall, our projection model has $68\times$ fewer parameters and it is $280$-$853\times$ faster in latency compared to simply decoding, upscaling and then encoding.

**Upscaling Stage.** Lastly, in the *upscaling* stage, *MobilePicasso* uses the super-resolution model (Noroozi et al., 2024) on the upscaled latents to generate high-resolution images (4K). Moreover, in this stage, *MobilePicasso* integrates our proposed *adaptive context-preserving tiling (ACPT)* and *model/system co-design* to minimise the latency for generating high-resolution images.

In summary, our 3-stage hybrid pipeline achieves fully on-device image editing for high-resolution images with significantly improved image quality and faster inference.

## 3.2 HALLUCINATION-AWARE LOSS

Prior image editing works have contributed to enhancing the visual quality, text alignment, and fidelity of edited images (Zhang et al., 2023a; Wasserman et al., 2024). However, existing image editing approaches often fail by producing hallucinations even at a standard resolution ($512 \times 512$). For instance, Figure 5 presents failure cases[1] of prior works such as IP2P, MagicBrush, and PIPE. The previous SOTA image editing approach, PIPE, enhanced image quality metrics; however, PIPE is still limited in its use cases as it primarily focuses on the addition type of prompts (Wasserman et al., 2024).

**Hallucination-Aware Loss.** We propose *hallucination-aware loss* to mitigate hallucinations by the image editing system at a standard resolution ($512 \times 512$). Hallucination-aware loss captures the amount of the hallucinated area of the edited images and penalises the model to reduce such hallucination. Specifically, we employ the hallucination detection model (Zhang et al., 2023b) that identifies the unrealistic artefacts within a given image. We then compute the area of artefacts and add it as an additional loss term in the diffusion loss, which acts as a regulariser penalising hallucinations

---

[1]These examples were chosen to show some typical situations we came across, although we do not include every possible case.

within an edited image. Formally, our hallucination-aware loss $L$ is computed as follows:

$$L_{LDM} = \mathbb{E}_{\boldsymbol{\epsilon}\sim\mathcal{N}(0,1),t,y}\left[\|\boldsymbol{\epsilon} - \boldsymbol{\epsilon_\theta}(\boldsymbol{z}_t, t, \boldsymbol{c}_\mathrm{I}, \boldsymbol{c}_\mathrm{T}))\|_2^2\right]$$
$$L_{Hallu} = \mathbb{E}_{\boldsymbol{\epsilon}\sim\mathcal{N}(0,1),t,y}\left[\|\mathcal{M}(\mathcal{D}(\boldsymbol{g_\theta}(\boldsymbol{z}_t, t, \boldsymbol{c}_\mathrm{I}, \boldsymbol{c}_\mathrm{T})))\|_2^2\right] \quad (3)$$
$$L = L_{LDM} + \lambda * L_{Hallu}$$

where $y = (\boldsymbol{x}, \boldsymbol{c}_\mathrm{I}, \boldsymbol{c}_\mathrm{T})$ is a triplet of target image, input image, and the edit prompt, and $\mathcal{M}$ is the hallucination detection model providing the degree of hallucination in a given image (Zhang et al., 2023b), and $\lambda$ controls the dominance of the hallucination-aware loss term.

**Hallucination-based Data Filtering.** Additionally, we observed that many images from the IP2P dataset, used for training the image editing models such as IP2P and MagicBrush, contain artefacts, similar to the findings in prior work (Liang et al., 2024). This allows the image editing models to generate implausible parts or objects. Therefore, we filter out images that exceed a certain threshold of artefact ratio, to improve the overall image editing dataset quality. We employ the hallucination detection model (Zhang et al., 2023b) to assess the ratio of artefacts in the image editing datasets and then filter out images that exceed a manually set threshold. In our main experiments, we set a threshold, filtering out around 15% of the training data.

### 3.3 ADAPTIVE CONTEXT-PRESERVING TILING

Our 3-stage hybrid pipeline largely alleviates the hallucination problem that occurs when directly generating high-resolution images solely based on image-editing models. Applying a tiling-based super-resolution model further requires considerable computational overheads due to the tiling overlaps. For instance, 0% overlap (Figure 3a) causes glitches/seams between neighbouring tiles as illustrated in Figure 6 (bottom row, third column).

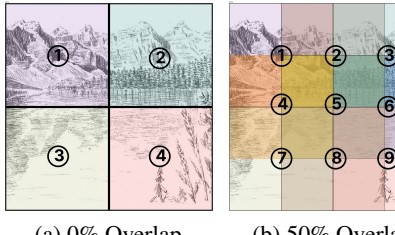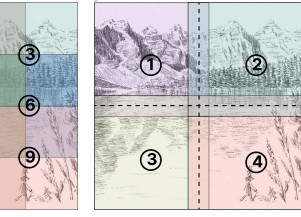

(a) 0% Overlap     (b) 50% Overlap     (c) Adjacent Padding

Figure 3: The on-device tiling strategy with different overlap ratios and our proposed adjacent padding with 0% overlap.

Hence, we propose *Adaptive Context-Preserving Tiling (ACPT)* to solve the tiling issue and improve the end-to-end latency. ACPT consists of two components: (1) adjacent padding and (2) adaptive tiling overlap mechanism.

**Adjacent Padding.** In practice, the default padding strategies are often zero padding which fills padding with zero values or reflect padding which repeats the tile itself in the padding (von Platen et al., 2022). However, these strategies result in image quality degradation as they lack information about neighbouring tiles. Therefore, we propose a new padding strategy, *adjacent padding*, that uses surrounding tiles as padding so that it ensures that padding is naturally smooth and consistent across neighbouring tiles (see Figure 3c). As adjacent padding provides contextual information from neighbouring tiles, super-resolution models can generate smooth images consistent across connected tiles. Notably, our padding strategy enables 0% tile overlap without image quality degradation (see Figure 6) yet negligible memory and computation overheads (see Table 3) over 0% overlap, highlighting its impact on on-device efficiency.

**Adaptive Tiling Overlap Mechanism.** We develop an *adaptive tiling overlap mechanism* to further optimise the execution of *MobilePicasso* for diverse input resolutions by leveraging adjacent padding and tile overlap adaptively. We provide the detailed ACPT algorithm (see Algorithms 1 and 2). The adaptive strategy selection (Lines 2-21 in Algorithm 1) optimises processing approach based on input characteristics. Specifically, ACPT relies on adjacent padding when input resolution is a multiple of tile size; otherwise, it executes with small overlap ratios between tiles without any padding. Our adaptive mechanism automatically selects the strategy based on input resolutions, ensuring rapid end-to-end processing across diverse input resolutions, while making *MobilePicasso* highly versatile for real-world applications (see Table 5 for ACPT's effectiveness using super-resolution quality metrics). Furthermore, by incorporating contextual information from adjacent tiles, ACPT effectively eliminates visible seams and artefacts commonly occurring when 0% tile overlap or other tiling approaches are used (see Figure 6).

### 3.4 MODEL-SYSTEM CO-DESIGN

To further optimise the computational efficiency of high-resolution I2I editing, we propose a *model/system co-design* approach that optimises on-device resource overheads. First, we conduct an extensive empirical analysis of the operational efficiency (latency) of DMs. Our analysis reveals compelling insights into the relationship between these factors and system performance. That is, in multi-tile scenarios, there exist a non-linear relationship where optimal performance is achieved with intermediate tile sizes ranging from 400 to 600 pixels. Notably, our finding challenges the conventional assumption that end-to-end latency consistently increases with tile size when processing a tile, as shown in Figure 4.

This counter-intuitive result stems from the complex interplay between several factors. Firstly, smaller tiles (e.g., 128 pixels) require more inference passes and an increased amount of additional computations due to tiling overlap. Secondly, larger tiles (e.g., 768 pixels) strain device memory and processing capabilities. Finally, intermediate tile sizes (e.g., 384 and 512 pixels) balance these competing factors while maximising NPU utilisation. Note that the performance improvements are substantial: our identified optimal tile size configura-

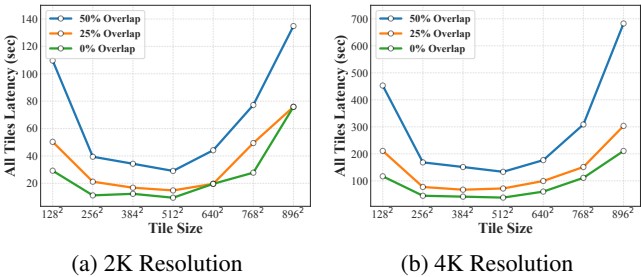

(a) 2K Resolution  (b) 4K Resolution

Figure 4: Figures (a,b) show latency results of the tiling-based approach for high-resolution images according to different tile sizes and overlap ratios. All measured on the NPU of the Snapdragon 8 Gen 2 chipset.

tion achieves a 3-4× and 4-8× latency reduction compared to smaller (128 pixels) and larger tiles (896 pixels), respectively. We leverage these insights in our model/system co-design, tuning our DMs for image editing and upscaling to operate efficiently with these identified optimal tile sizes, while operating within mobile memory limitations.

## 4 EVALUATION

### 4.1 EXPERIMENTAL SETUP

**Datasets:** We use IP2P (Brooks et al., 2022) and PIPE (Wasserman et al., 2024) datasets for finetuning the pretrained diffusion model (Rombach et al., 2022). The IP2P dataset contains synthetically generated source and target images with an edit prompt, and the PIPE dataset contains real target images with synthetic source images generated by filling the masked area with an inpainting model. We also use the MagicBrush (Zhang et al., 2023a) dataset to present quantitative evaluation results following (Wasserman et al., 2024). In addition, we employ the LIU4K dataset (Liu et al., 2020) consisting of high-resolution images to further evaluate our method.

**Architectures:** For the image editing stage, we employ SDv1.5 (Rombach et al., 2022) following prior works (Brooks et al., 2022). For the latent projection stage, we use a tiny autoencoder (Ollin Boer Bohan, 2024) and train lightweight convolutional networks to perform projections in the latent space. For the upscaling stage, we use diffusion-based super-resolution model (Noroozi et al., 2024) to leverage its efficient upscaling capability.

**Evaluation:** To evaluate the performance of *MobilePicasso*, we construct the *MobilePicasso test set* using high-resolution images from LIU4K (Liu et al., 2020) consisting of five categories such as (1) animals, (2) architecture, (3) city, (4) food, and (5) landscape, as well as edit prompts. We developed edit instructions that range from simple attribute modifications to complex style transformations, accounting for 1,000 text-image pairs. We then present qualitative results with edited images using instruction-based DMs on our *MobilePicasso* test set to evaluate visual quality and text alignment. Following Zhang et al. (2023a); Wasserman et al. (2024); Sheynin et al. (2023), we employ various metrics such as L1, L2, CLIP-I, DINO, CLIP-T, and FID. On the system front, we present the latency and memory usage of running *MobilePicasso* on-device. Also, our user study evaluates each method regarding four aspects: (1) Text Alignment: How well a generated image follows the description of the text prompt, (2) Image Consistency: How closely the objects in the generated image match those in the reference images, (3) Resilience to Hallucination: Whether there exist hallucinations in

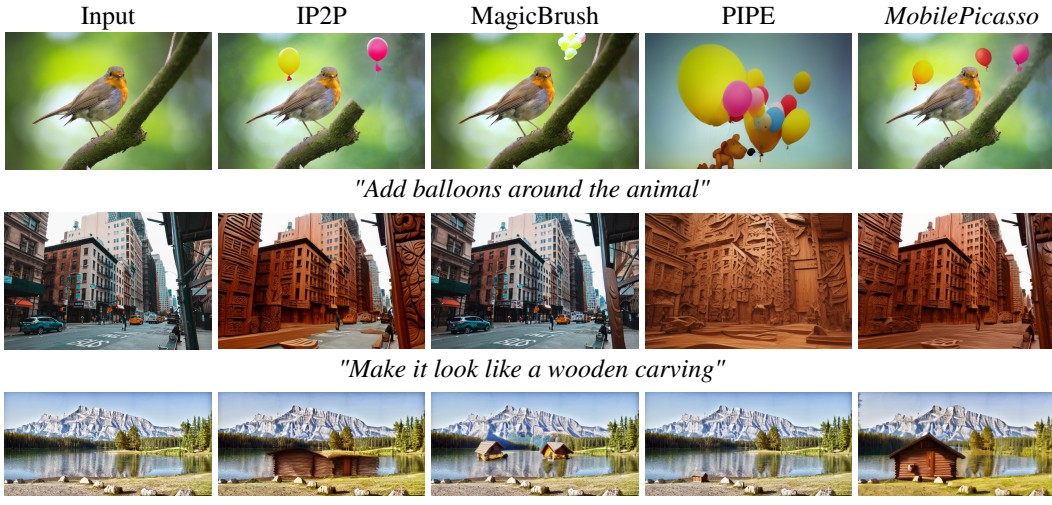

| Input | IP2P | MagicBrush | PIPE | *MobilePicasso* |

*"Add balloons around the animal"*

*"Make it look like a wooden carving"*

*"Add a cosy cabin made of wood"*

Figure 5: Qualitative comparison among image editing models and *MobilePicasso* given images at standard resolutions (e.g., $512 \times 512$).

the generated images, and (4) Image Quality: The overall visual quality considering clarity, colour, composition, and other factors.

**Baselines:** Regarding image quality, we compare *MobilePicasso* with the following image editing baselines: (1) IP2P (Brooks et al., 2022), (2) MagicBrush (Zhang et al., 2023a), and (3) PIPE (Wasserman et al., 2024). In terms of latency, we compare *MobilePicasso* with baselines using (1) direct high-resolution image editing on a GPU and (2) various tiling overlaps (e.g., 0%, 25%, and 50%).

Table 1: Performance of image-editing baselines based on the MagicBrush test set.

| Method | L1 ↓ | L2 ↓ | CLIP-I ↑ | DINO ↑ | CLIP-T ↑ | FID ↓ |
|---|---|---|---|---|---|---|
| IP2P | 0.100 | 0.031 | 0.897 | 0.725 | 0.269 | 52.8 |
| MagicBrush | 0.077 | 0.028 | 0.934 | 0.843 | **0.274** | 35.8 |
| PIPE | 0.072 | 0.025 | 0.934 | 0.820 | 0.269 | 78.4 |
| Ours (Data Filtering) | 0.072 | 0.023 | 0.938 | 0.840 | 0.269 | 35.6 |
| Ours (+ Hallucination-aware Loss) | **0.069** | **0.021** | **0.940** | **0.845** | 0.270 | **34.1** |

## 4.2 MAIN RESULTS

**Image Quality at Standard Resolutions:** Figure 5 shows example outputs from *MobilePicasso* and the baselines. IP2P is observed to often generate hallucinated images, this is due to their dataset containing hallucinated images derived from the synthetic data generation process. MagicBrush was fine-tuned using the MagicBrush dataset, consisting of authentic images. However, on the *MobilePicasso* test set, its output images have even more hallucinations than IP2P due to its limited diversity of images and prompts (MagicBrush training data is only 10K image-text pairs), similar to prior work (Sheynin et al., 2023). PIPE trained on both IP2P and PIPE datasets, which contain around 800K realistic image-edit prompt pairs, improves the visual quality over IP2P and MagicBrush. *MobilePicasso* shows qualitatively better visual quality, outperforming all the baselines.

Following Zhang et al. (2023a); Wasserman et al. (2024); Sheynin et al. (2023), we perform another quantitative evaluation based on the MagicBrush test set. Table 1 shows that *MobilePicasso* achieves the best scores for the image editing task, further demonstrating the superiority of our method. *These results demonstrate the effectiveness of our proposed hallucination loss and artefact filtering process.*

**User Study:** To provide more compelling and robust statistical evidence regarding hallucination reduction, we conducted an extended user study with 46 participants, and each par-

Table 2: User study on perceived quality of edited images regarding four aspects with mean (standard error) for three baselines and *MobilePicasso*.

| Methods | Text Alignment | Image Consistency | Resilience to Hallucination | Image Quality |
|---|---|---|---|---|
| IP2P | 3.65 (±0.15) | 4.18 (±0.12) | 3.76 (±0.14) | 3.73 (±0.14) |
| MagicBrush | 2.89 (±0.19) | 2.98 (±0.20) | 2.84 (±0.19) | 3.04 (±0.16) |
| PIPE | 2.90 (±0.18) | 3.20 (±0.20) | 2.98 (±0.20) | 2.98 (±0.18) |
| *MobilePicasso* | **4.45** (±0.09) | **4.39** (±0.09) | **4.29** (±0.10) | **4.40** (±0.09) |

ticipant rates four methods with four metrics across eight evaluation sets, resulting in 5,888 user-

| Input | IP2P (w/o Tiling) | IP2P (Tiling, 25/0% OL) | Ours (ACPT, 0% OL) |

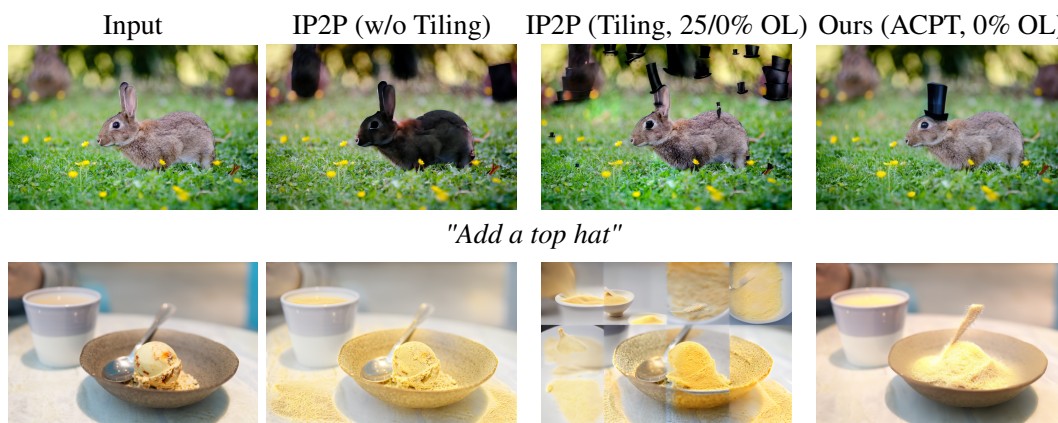

*"Add a top hat"*

*"Spread parmesan cheese on top"*

Figure 6: Qualitative comparison among instruction-based image editing models and *MobilePicasso* for high-resolution input images (2K+). (1) Note that the baselines are based on IP2P without the tiling approach to simulate the server-based image editing which serves as the upper bound. (2) The other baselines are IP2P with tiling using different tiling ratios (i.e., 25% for top row, 0% for bottom row) which serves as strong baselines with similar resource constraints to *MobilePicasso*. OL indicates an overlap between neighbouring tiles.

perceived ratings (see Appendices E and F.1 for study design and domain-specific results, respectively). Table 2 shows that *MobilePicasso* substantially improves Text Alignment (by 22-54%), Robustness to Hallucination (by 14-51%), and Image Quality (by 18-48%) compared to all the baselines except for Image Consistency (by 5-47%) where IP2P shows similar performance to *MobilePicasso*. *These human evaluation results provide much stronger evidence for our claims regarding hallucination reduction, thereby improving overall image quality, than automated metrics alone.*

**Image Quality at High Resolutions:**
Figure 6 shows the qualitative comparison of IP2P baselines with and without the tiling approach for high-resolution image editing. These baselines do not use our 3-stage hybrid pipeline but apply image editing directly to high-resolution images without subsequent upscaling procedures. First, IP2P without tiling, server-based high-resolution image editing, which simulates that it is free from the on-device memory constraint, often

Table 3: Comparison of the end-to-end latency and memory usage for editing 4K-resolution images. Note baseline (w/o tiling) crashes due to OOM on-device and thus we measured its latency and memory on a GPU simulating a server-based inference scenario.

| Device | Method | Latency | Ratio | Memory | Ratio |
|--------|--------|---------|-------|--------|-------|
| GPU | Baseline (w/o Tiling) | 197s | 4.71× | 76.2 GB | **71.9×** |
| Phone | Baseline (w/ Tiling, 50% OL) | 2,340s | **55.8×** | 1.06 GB | 1× |
| Phone | Baseline (w/ Tiling, 25% OL) | 1,201s | 28.6× | 1.06 GB | 1× |
| Phone | MobilePicasso | **42.0s** | 1× | 1.15 GB | 1.09× |

produces images that do not follow the edit prompt well. Second, IP2P with tiling of 25% or 0% overlaps produces much more unrealistic artefacts and glitches/seams between neighbouring tiles, drastically degrading the overall image quality. *Yet, MobilePicasso's 3-stage hybrid pipeline equipped with the proposed hallucination-aware loss and ACPT, significantly improves the image quality by reducing unrealistic artefacts and enhancing prompt alignment while reducing the end-to-end latency, as presented next.*

**End-to-end Latency:** We now examine the effectiveness of our ACPT by presenting the end-to-end latency results of *MobilePicasso* with our proposed ACPT, server-based baseline without tiling, on-device baseline with tiling (25%, 50% overlaps). Table 3 demonstrates that *MobilePicasso* achieves up to 55.8× reduction in end-to-end latency compared to the baselines while achieving better image quality. *MobilePicasso* achieves up to 55.8× speed-ups over on-device baselines with 50% overlap. Interestingly, despite on-device resource constraints of Snapdragon 8 Gen 2 NPU (on Galaxy S23), *MobilePicasso*'s processing time for high-resolution image editing is even faster than server-based baselines on a powerful GPU (NVIDIA A100) by a large margin (4.71×). *This demonstrate the effectiveness of MobilePicasso's 3-stage hybrid pipeline and ACPT.*

**Peak Memory:** We investigate the peak memory usage of *MobilePicasso* and the baselines as shown in Table 3. To begin with, directly applying image editing on high-resolution images (i.e., baselines without tiling) crashes due to OOM, and thus we report the peak memory consumed on a GPU for this baseline (76.2 GB). The result demonstrates that on-device baselines with tiling and *MobilePicasso* consume significantly smaller memory of 1.06-1.15 GB. *MobilePicasso* incurs a slight memory increase by 9% (peaked at 1.15 GB). Yet, its peak memory is well under the on-device memory constraint of the mobile NPU (e.g., 2 GB).

## 5 ABLATION STUDY AND ANALYSIS

**Latency Breakdown Analysis of Hybrid Pipeline:** Table 4 provides a latency breakdown analysis revealing each MobilePicasso component's contribution. ACPT optimisation (3.12× speedup) eliminates redundant pixel processing from traditional 50% overlap while maintaining visual quality through adjacent padding. Hybrid

Table 4: Breakdown of the end-to-end latency and memory usage for editing 4K-resolution images. OL indicates an overlap ratio between neighbouring tiles.

| Device | Method | Latency | Ratio | Memory | Ratio |
|---|---|---|---|---|---|
| Phone | Baseline (w/ Tiling, 50% OL) | 2,340s | 1× | 1.06 GB | 1× |
| Phone | + ACPT, 0% OL | 749s | **3.12×** | 1.15 GB | 1.09× |
| Phone | + Hybrid Pipeline | 48.8s | **15.35×** | 1.15 GB | 1.09× |
| Phone | + Learnable Latent Projection | 42.0s | **1.16×** | 1.15 GB | 1.09× |

Pipeline Integration (15.35× speedup) reveals that most significant gain comes from performing editing at $512 \times 512$ resolution, avoiding quadratic scaling with image dimensions. Learnable Latent Projection (1.16× speedup) replaces expensive VAE encoding/decoding operations with a single learned transformation in latent space. The multiplicative relationship ($3.12 \times 15.35 \times 1.16 = 55.8$) demonstrates that our optimisations address different computational bottlenecks, validating our systematic decomposition approach.

**ACPT Effectiveness Analysis:** We quantify ACPT's effectiveness using super-resolution quality metrics, PSNR and SSIM, that capture both global coherence and local detail preservation (Zhang et al., 2021; Noroozi et al., 2024; 2025). Table 5 shows that our ACPT approach achieves 19.73 dB PSNR, outperform-

Table 5: Comparison between ACPT and baselines with various tiling overlaps on 4K-resolution images from the LIU4K dataset. OL indicates a tile overlap.

| Metric | 50% OL | 25% OL | 0% OL | 0% OL Reflect Padding | ACPT, 0% OL Adjacent Padding |
|---|---|---|---|---|---|
| PSNR | **19.78** | 19.63 | 19.34 | 18.82 | **19.73** |
| SSIM | **0.5470** | 0.5386 | 0.5277 | 0.5150 | **0.5389** |

ing 0% overlap baseline (19.34 dB) and representing 99.7% of the upper bound quality (19.78 dB) based on 50% overlap with significantly lower computational overhead. For SSIM, ACPT achieves 98.5% of upper bound performance (0.5389 vs 0.5470), demonstrating excellent preservation of structural information and perceptual quality. Moreover, we demonstrate that ACPT is applicable across both diffusion-based and GAN-based super-resolution architectures (see Appendix F.4). Figure 8 compares padding strategies qualitatively. Reflect padding produces visible glitches between neighbouring tiles, whereas our adjacent padding provides seamless transitions by using actual neighbouring image content as contextual information.

**Hallucination-Aware Loss Ablation:** We systematically evaluate our hallucination-aware loss and data filtering contributions. Table 1 shows that data filtering improves 17.8% on average across 6 quantitative metrics compared to IP2P. Also, hallucination-aware loss further improves 2.2% on average, with their combination achieving 20.0% overall improvement on average across six metrics. These results demonstrate that both components are necessary and complementary.

## 6 CONCLUSION

We have developed the first realistic high-resolution (4K) image editing framework on mobile devices, *MobilePicasso*, addressing practical challenges of hallucination, memory and compute constraints, and tiling issues. *MobilePicasso* integrates the 3-stage hybrid pipeline, hallucination-aware loss, adaptive tiling with adjacent padding, and model/system co-design. As a result, *MobilePicasso* produces high-resolution images with better visual quality with drastically smaller memory and compute costs on mobile devices than all the prior works requiring high-end GPUs.

In the future, we will extend beyond the immediate application of image editing to other generative AI tasks such as text/sketch-to-image generation, style transfer, and image inpainting/outpainting.

ETHICS STATEMENT

The widespread adoption of cloud-based image editing solutions has raised significant societal concerns regarding privacy. Such systems require users to upload personal photos to remote servers, creating privacy vulnerabilities. Our work, *MobilePicasso*, directly addresses these societal concerns by enabling high-quality image editing directly on mobile devices and eliminating the need to transmit sensitive data to external servers. While on-device image editing technology could potentially enable misuse through easier access to image manipulation capabilities on personal devices, the democratisation of image editing tools also provides substantial positive benefits, including privacy protection, offline accessibility, and reduced computational infrastructure dependence.

Our user study involving 46 participants was conducted in accordance with standard research ethics practices. Participation was voluntary, and no personally identifiable information was collected. All participants were informed about the study's purpose and provided consent for their participation. Furthermore, this work does not involve sensitive personal data beyond the voluntary user study. The datasets used in our experiments (IP2P, PIPE, MagicBrush, LIU4K) are publicly available benchmarks that have been widely adopted in the research community without ethical concerns. Our mobile device testing was conducted on commercially available hardware without accessing private user data. We believe the contributions of this paper align with the ICLR Code of Ethics.

REPRODUCIBILITY STATEMENT

We have taken several steps to ensure the reproducibility of our results. All datasets employed in this study are publicly available and described in the main text and appendix. Detailed descriptions of model architectures, hyperparameters, and training protocols are provided in the paper and appendix. Furthermore, we provide comprehensive algorithmic specifications (including pseudocode of Algorithms 1 and 2 for ACPT) and implementation details in the appendix. Our 46-participant user study methodology is fully documented in Appendix E to enable replication of the evaluation protocol. The complete source code will be made available upon paper acceptance.

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

# Supplementary Material
## On-Device High-Resolution Image Editing with Hallucination-Aware Loss and Adaptive Tiling

## Table of Contents

## A  DETAILED EXPERIMENTAL SETUP

### A.1  DATASETS

We use IP2P (Brooks et al., 2022) and PIPE (Wasserman et al., 2024) datasets for finetuning our pre-trained diffusion model (Rombach et al., 2022). The IP2P dataset contains around 300,000 synthetic source-target image pairs with edit instructions, covering local modifications, object transformations, and style transfer. The PIPE dataset is an even larger dataset with 1M text-image pairs that combine real target images with synthetically generated source images through the inpainting pipeline.

For evaluation purposes, we employ two additional datasets. The MagicBrush (Zhang et al., 2023a) dataset serves as our primary quantitative benchmark, following established protocols in Wasserman et al. (2024). This dataset includes 1,000 diverse test cases with carefully curated edit instructions and ground-truth edited images for comparison. The standardized evaluation protocols enable consistent measurement of editing quality and fidelity across different methods. The LIU4K dataset (Liu et al., 2020) contains 4,000 images at various resolutions such as 4K or 6K. These images encompass diverse photographic styles and subjects, maintaining professional-grade image quality across multiple lighting conditions and scenes. The high resolution and variety make this dataset particularly valuable for evaluating the real-world applicability of our method.

### A.2  ARCHITECTURES

For the image editing stage, we employ SD1.5 (Rombach et al., 2022) following prior works Brooks et al. (2022). The model uses a UNet backbone with 860M parameters, operating at a standard resolution of $512 \times 512$.

Moreover, for the latent projection stage, we use tiny autoencoder (Ollin Boer Bohan, 2024) and lightweight convolutional neural networks (CNNs) to perform projection in latent space. We train our lightweight CNNs to learn a mapping between the processed latent from the image edit stage and the encoded latent from the upscale stage.

Finally, for the upscale stage, we utilise a diffusion-based model (Noroozi et al., 2024) that achieves upscaling through a UNet architecture. The model is optimised for both quality and computational efficiency at inference time.

### A.3  EVALUATION

***MobilePicasso* Test Set.** To evaluate *MobilePicasso*, we construct the *MobilePicasso test set* using high-resolution images from LIU4K (Liu et al., 2020) across five categories:

- **Animals**: The animal category includes various species photographed in various poses, lighting conditions, and natural habitats, ensuring robust evaluation of fine detail preservation in fur and feathers, etc.
- **Architecture**: The architecture category include historical monuments, modern buildings, and intricate architectural details.
- **City**: The city category captures urban environments across different times of day, weather conditions, and complex scene compositions.
- **Food**: The food category presents culinary items with varying textures, colours, and presentation styles.
- **Landscape**: The landscape category encompasses natural scenery across different seasons, weather conditions, and lighting scenarios.

For each category in our test set, we developed a comprehensive set of edit instructions that range from simple attribute modifications to complex style transformations. These instructions are carefully crafted to evaluate different aspects of the editing process: general edits with local modifications test precise control and style edits for detail preservation and global adjustments assess consistency in style application. In total, our *MobilePicasso* test set accounts for 1,000 text-image pairs. This diverse instruction set enables a thorough assessment of both editing capability and generalization performance. Our quantitative evaluation framework employs multiple complementary metrics to

assess different aspects of editing quality. Also, our *MobilePicasso* test set is used for qualitative evaluation with visual inspection.

**Hallucination Metric.** We also measure the hallucination metric. We utilise partial artefacts ratios (PAR), representing the unrealistic portion of an image (Zhang et al., 2023b), as the hallucination metric. PAR indicates undesired artefacts or modifications, providing crucial insights into editing reliability. We perform a quantitative evaluation using the hallucination metric on the *MobilePicasso* test set.

**Quantitative Metric.** Following established protocols from previous works (Zhang et al., 2023a; Wasserman et al., 2024; Sheynin et al., 2023), we implement a comprehensive suite of evaluation metrics. The L1 and L2 metrics provide pixel-level accuracy measurements, particularly valuable for assessing local editing precision. CLIP-I scores evaluate perceptual quality, measuring how well the edited images align with human visual expectations. DINO features assess semantic consistency, ensuring that edited images maintain appropriate high-level semantic relationships. CLIP-T scores quantify text-image alignment, measuring how accurately the edited images reflect the given instructions.

**System Measurements.** On the system performance front, we conduct extensive measurements of computational efficiency. Latency measurements capture end-to-end processing time. The memory footprint is measured throughout our hybrid pipeline, with particular attention to peak memory usage during different processing stages. Note that memory requirement becomes a key bottleneck in deploying large-scale DMs as it often incurs out-of-memory errors due to the excessive memory requirements of DMs. Hence, system performance measurements provide valuable insights into real-world deployment scenarios.

### A.4 BASELINES

Regarding image quality, we compare *MobilePicasso* with the following image editing baselines: (1) IP2P (Brooks et al., 2022), (2) MagicBrush (Zhang et al., 2023a), and (3) PIPE (Wasserman et al., 2024). In terms of latency, we compare *MobilePicasso* with IP2P using (1) direct high-resolution image editing on a GPU and (2) various tiling overlaps (e.g., 0%, 25%, and 50%).

## B DETAILS OF LEARNABLE LATENT PROJECTION

### B.1 DESIGN RATIONALE

Our learned projection layer effectively upscales the edited latent from standard resolution (512x512) to high resolutions (2K/4K) without image distortion, whereas linear projections, such as bilinear, bicubic, all lead to significant image distortion when upscaling from 512 to 2K. Furthermore, the learnable projection layer enables avoiding frequent execution of the VAE decoder and encoder, which incur high memory usage and latency.

### B.2 TRAINING DETAILS

We used the MS-COCO dataset as a training dataset to train the learnable latent projection component of our 3-stage hybrid pipeline. Our latent projection model includes Tiny AutoEncoder which is frozen during training and a lightweight projection model consisting of multiple blocks of convolutional layers which is trained. We experimented with different numbers of convolutional blocks, [3, 6, 12] and widths of each block, [16, 32, 64, 128]. We used ADAM optimiser and mean-squared error loss. We trained up to 10 epochs, yet we found that the performance converges quickly after 2 or 3 epochs (around 10 hours on a single A100 GPU).

## C ALGORITHM SPECIFICATION OF ACPT

In this section, we provide the detailed algorithm specification of Adaptive Context-Preserving Tiling (ACPT) (see Algorithms 1 and 2), enabling memory-efficient high-resolution image synthesis through intelligent adjacent padding and adaptive tiling.

**Design Rationale:** The ACPT algorithm addresses the fundamental trade-off between computational efficiency and visual quality in high-resolution image processing. Traditional tiling with overlap ratio $r$ requires processing $(1/(1-r))^2$ times more pixels. For 50% overlap, this represents $4\times$ overhead, making mobile deployment impractical.

Our adjacent padding innovation provides contextual information with minimal computational overhead by using actual neighbouring image content, providing natural contextual transitions that super-resolution models leverage effectively. This approach effectively alleviates the issue of artificial seams/glitches, while maintaining 0% overlap efficiency. Note that traditional padding strategies (e.g., reflection) create artificial glitches as demonstrated in Figure 8 qualitatively and in Table 5 quantitatively.

The adaptive strategy selection (Lines 2-21 in Algorithm 1) optimises processing approach based on input characteristics. When image dimensions are perfectly divisible by tile size, the algorithm employs adjacent padding with 0% overlap for maximum efficiency. For non-divisible dimensions, it gracefully degrades to a small overlap strategy, ensuring consistent quality across arbitrary input resolutions while minimising computational overhead.

---

**Algorithm 1:** Adaptive Context-Preserving Tiling (ACPT)

---

**Input:** $image$, $model$, $tile\_size$, $padding\_size$, $overlap\_ratio$
**Output:** $output\_image$

1  $height, width \leftarrow image.shape[:2]$;
2  $use\_adjacent\_padding \leftarrow (width \bmod tile\_size = 0) \wedge (height \bmod tile\_size = 0)$;
3  $output\_image \leftarrow \textbf{zeros\_like}(image)$;
4  **if** $use\_adjacent\_padding$ **then**
    // Strategy A: Adjacent padding with 0% overlap
5      $num\_tiles\_x, num\_tiles\_y \leftarrow \lfloor width/tile\_size \rfloor, \lfloor height/tile\_size \rfloor$;
6      **for** $tile\_y \leftarrow 0$ **to** $num\_tiles\_y - 1$ **do**
7          **for** $tile\_x \leftarrow 0$ **to** $num\_tiles\_x - 1$ **do**
            // Extract tile with adjacent padding
8              $padded\_tile \leftarrow \text{ExtractWithAdjacentPadding}(image, tile\_x \times tile\_size, tile\_y \times tile\_size, tile\_size, padding\_size)$;
            // Process and place in output
9              $processed\_tile \leftarrow model(padded\_tile)$;
10             $core\_region \leftarrow \text{RemovePadding}(processed\_tile, padding\_size)$;
11             $output\_image[tile\_y \times tile\_size : (tile\_y + 1) \times tile\_size, tile\_x \times tile\_size : (tile\_x + 1) \times tile\_size] \leftarrow core\_region$;

12 **else**
    // Strategy B: Small overlap for non-divisible dimensions
13     $overlap\_pixels \leftarrow \lfloor overlap\_ratio \times tile\_size \rfloor$;
14     $stride \leftarrow tile\_size - overlap\_pixels$;
    // Calculate tile positions and process with blending
15     $tile\_positions\_x \leftarrow \{0, stride, 2 \times stride, \ldots\} \cap [0, width - tile\_size]$;
16     $tile\_positions\_y \leftarrow \{0, stride, 2 \times stride, \ldots\} \cap [0, height - tile\_size]$;
17     **for** $tile\_y\_pos \in tile\_positions\_y$ **do**
18         **for** $tile\_x\_pos \in tile\_positions\_x$ **do**
19             $tile \leftarrow image[tile\_y\_pos : tile\_y\_pos + tile\_size, tile\_x\_pos : tile\_x\_pos + tile\_size]$;
20             $processed\_tile \leftarrow model(tile)$;
21             $\text{BlendTilesToOutput}(output\_image,$
22             $processed\_tile, tile\_x\_pos, tile\_y\_pos)$;

23 **return** $output\_image$

---

---

**Algorithm 2:** ExtractWithAdjacentPadding

---

**Input:** $image$, $start\_x$, $start\_y$, $tile\_size$, $padding\_size$
**Output:** $padded\_tile$
// Extract tile with adjacent padding from neighboring regions
1 $height, width \leftarrow image.shape[:2]$;
2 $padded\_size \leftarrow tile\_size + 2 \times padding\_size$;
3 $padded\_tile \leftarrow \mathbf{zeros}(padded\_size, padded\_size, image.shape[2])$;
 // Extract core tile
4 $core\_tile \leftarrow image[start\_y : start\_y + tile\_size, start\_x : start\_x + tile\_size]$;
5 $padded\_tile[padding\_size : padding\_size + tile\_size, padding\_size :$
 $padding\_size + tile\_size] \leftarrow core\_tile$;
 // Add adjacent padding from neighboring tiles
6 **if** $start\_y \geq padding\_size$ **then**
7  $\quad top\_pad \leftarrow image[start\_y - padding\_size : start\_y, start\_x : start\_x + tile\_size]$;
8  $\quad padded\_tile[0 : padding\_size, padding\_size : padding\_size + tile\_size] \leftarrow top\_pad$;
9 **if** $start\_x \geq padding\_size$ **then**
10  $\quad left\_pad \leftarrow image[start\_y : start\_y + tile\_size, start\_x - padding\_size : start\_x]$;
11  $\quad padded\_tile[padding\_size : padding\_size + tile\_size, 0 : padding\_size] \leftarrow left\_pad$;
 // Similar operations for bottom and right padding...
12 **return** $padded\_tile$

---

## D    Details of Handling High-Resolution Images

In this section, we describe the procedure of handling high-resolution images in our 3-stage hybrid pipeline. For example, when a 2K/4K image is given as input, we perform preprocessing using Lanczos resampling to downsample the input image so that at least one spatial dimension to a standard resolution (512). If an input is provided at a standard resolution, then we use it directly to *MobilePicasso*. Then, our system performs the following three stages:

Firstly, it performs editing at a standard resolution using the image-editing model fine-tuned with our hallucination-aware loss, Secondly, it applies projection in the latent space from edited latent to 2K/4K latent space using our learnable latent projection layer, finally, it performs super-resolution on the 2K/4K latent with ACPT and adjacent padding.

Note that direct editing at 4K resolution is fundamentally impossible on mobile devices. As shown in Table 3, running the baseline architecture IP2P without tiling requires 76.2 GB of memory, far exceeding mobile capabilities. Our 3-stage decomposition makes high-resolution editing tractable within mobile constraints.

## E    Detailed User Study Design

### E.1    Study Objectives and Methodology

Our user study validates *MobilePicasso*'s effectiveness in reducing hallucinations and improving image quality through human perceptual evaluation. While automated metrics provide quantitative assessments, human evaluation serves as the golden standard for measuring end-user experience and perceptual quality that automated quality metrics ultimately aim to optimise.

The study tests two primary hypotheses: (1) *MobilePicasso* generates images with significantly fewer hallucinations compared to baseline methods, (2) *MobilePicasso* produces images with better text alignment and overall visual quality. Additionally, we validate correlations between automated metrics and human perception.

**Participants:** We recruited 46 participants through university networks and online platforms to evaluate the perceptual quality differences between *MobilePicasso* and baseline methods. Participants were asked to assess edited images across multiple quality dimensions without prior knowledge of the methods being evaluated.

Figure 7: The user study interface that describes the objectives, instructions, evaluation criteria, as well as example image samples.

## E.2 EXPERIMENTAL INTERFACE AND PROTOCOL

We conducted the evaluation using Google Forms to ensure accessibility and ease of participation. Each evaluation set presents participants with a reference image, an edit instruction, and four generated images from different methods. Each participant assesses 8 evaluation sets, accounting for 32 total tasks (8 per method: IP2P, MagicBrush, PIPE, *MobilePicasso*). We employ *MobilePicasso* Test Set as reference images (see Appendix A.3 for the dataset details). To ensure randomness and eliminate position bias, we shuffled the location of images between different evaluation sets, so that no method

consistently appeared in the same position across multiple tasks. Participants rate each image on four dimensions:

**Evaluation Dimensions:**

- **Text Alignment:** How well the image follows the edit instruction (0=Poorly following or not following edit instruction, 5=Perfectly following edit instruction).

- **Image Consistency:** How well original elements are preserved (0=Significantly altered, 5=Perfectly preserved).

- **Resilience to Hallucination:** Whether there exist artefacts/hallucinations in the generated images (0=Major artefacts, 5=No unrealistic elements).

- **Image Quality:** Overall visual appeal and quality (0=Poor quality, 5=Superior quality)

**Session Structure:** Participants completed the evaluation at their own pace, with clear instructions provided at the beginning. The evaluation included training examples to familiarise participants with the rating criteria, followed by the main evaluation tasks.

### E.3 RESULTS SUMMARY

As shown in Table 2, the substantial improvements in human-perceived quality across multiple dimensions demonstrate that our algorithmic innovations address real perceptual limitations rather than merely optimising automated metrics, providing strong evidence for *MobilePicasso*'s practical value and potential for real-world deployment. Figure 7 shows our user study interface.

Also, note that each participant of our user study rates four methods (three baselines and our work) in four metrics (Text Alignment, Image Consistency, Resilience to Hallucination, Image Quality) across 8 evaluation sets, accounting for 128 ratings. In sum, we collected 5,888 user-perceived ratings in our user study, which ensures the robust statistical significance of our results.

**Key Qualitative Findings:** Participants consistently noted that *MobilePicasso* images "look more natural and realistic" with "better integration of new elements" and "fewer weird artefacts and floating objects." These observations directly validate our technical objectives of reducing hallucinations and improving image quality through our novel architectural and training approaches. Additionally, when participants are asked such a question, "When editing personal photos, where would you prefer the processing to happen?", 69.6% of them prefer the image editing to be processed "entirely on their devices", 26.1% has "no preference", and only 4.3% favours the processing to happen "on cloud servers."

## F ADDITIONAL EXPERIMENTAL RESULTS

In this section, we present additional experimental results and analyses that are not included in the main content of the paper due to the page limit.

### F.1 DOMAIN-SPECIFIC USER STUDY RESULTS

Our *MobilePicasso* test set spans five diverse categories (animals, architecture, city, food, landscape) and edit instructions (from simple attribute modifications to complex style transformations), accounting for 1,000 text-image pairs in total, which covers major photographic domains. *This comprehensive coverage demonstrates MobilePicasso's domain generalisation capability within natural image distributions.*

We conducted a comprehensive cross-domain analysis showing consistent baseline gaps across all domains, as shown in Table 6. The consistent performance improvements across diverse domains indicate robust generalisation within our evaluation scope. Notably, our method maintains superiority even in challenging scenarios like city scenes with complex lighting and architectural details.

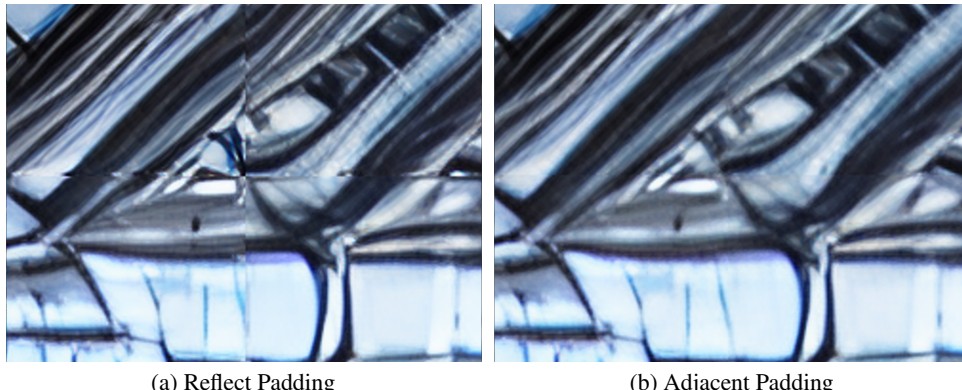

| (a) Reflect Padding | (b) Adjacent Padding |

Figure 8: Comparison between padding strategies: reflect padding and our proposed adjacent padding. 0% tiling overlap is used.

Table 6: Domain-specific user study results based on image quality metrics across five diverse categories (City, Animal, Landscape, Food, Architecture) using the *MobilePicasso* test set.

| Methods | City | Animal | Landscape | Food | Architecture |
|---|---|---|---|---|---|
| IP2P | 3.74 (±0.12) | 4.31 (±0.11) | 3.77 (±0.13) | 2.67 (±0.15) | 3.52 (±0.12) |
| MagicBrush | 3.29 (±0.15) | 2.55 (±0.18) | 3.03 (±0.19) | 3.64 (±0.15) | 3.17 (±0.19) |
| PIPE | 2.88 (±0.18) | 1.71 (±0.23) | 3.67 (±0.13) | 2.95 (±0.18) | 3.52 (±0.13) |
| *MobilePicasso* | **3.86** (±0.11) | **4.56** (±0.09) | **4.38** (±0.08) | **4.71** (±0.08) | **4.33** (±0.09) |

### F.2 DETAILED LATENCY BREAKDOWN ANALYSIS OF HYBRID PIPELINE

This subsection presents the detailed description of results shown in Table 4. It provides a latency breakdown analysis revealing the distinct performance contribution of each *MobilePicasso* component. *These results demonstrate the effectiveness of our proposed hybrid pipeline.*

**ACPT Optimisation (3.12× speedup):** Traditional 50% overlap requires processing more pixel data than necessary. Our ACPT eliminates this redundancy while maintaining visual quality through adjacent padding. The slight memory increase (9%) is negligible compared to the 3.12× speedup achieved.

**Hybrid Pipeline Integration (15.35× speedup):** The most significant gain comes from our computational decomposition through our hybrid pipeline. Direct high-resolution editing scales quadratically with image dimensions due to attention mechanisms in diffusion models. By performing editing at 512×512 resolution, we reduce computational complexity drastically.

**Learnable Latent Projection (1.16× speedup):** This component replaces expensive VAE decoding and bilinear upsampling, followed by VAE encoding with a single learned transformation. Maintaining latent representations throughout eliminates costly VAE encoding/decoding operations. The modest gain reflects the already optimised pipeline state. The multiplicative relationship (3.12 × 15.35 × 1.16 = 55.8) indicates that our optimisations address different computational bottlenecks, validating the systematic decomposition approach.

### F.3 ACPT EFFECTIVENESS QUANTIFICATION

In this subsection, we present further description of the effectiveness quantification of ACPT in alleviating tiling glitches using super-resolution quality metrics such as PSNR and SSIM that capture both global coherence and local detail preservation (Zhang et al., 2021; Noroozi et al., 2024; 2025). *These results reveal the effectiveness of our ACPT with adjacent padding during the super-resolution stage.*

Table 7: Comparison between ACPT and baselines with various tiling overlaps on 4K-resolution images from the LIU4K dataset. OL indicates a tile overlap. ACPT with adjacent padding demonstrates consistent improvements across three different architectures.

| Architecture | Metric | 50% OL | 25% OL | 0% OL | 0% OL Reflect Padding | ACPT, 0% OL Adjacent Padding |
|---|---|---|---|---|---|---|
| YONOS-SR (U-Net) | PSNR | **19.78** | 19.63 | 19.34 | 18.82 | **19.73** |
| YONOS-SR (U-Net) | SSIM | **0.5470** | 0.5386 | 0.5277 | 0.5150 | **0.5389** |
| Edge-SD-SR (U-Net) | PSNR | **20.12** | 20.06 | 19.85 | 19.75 | **20.14** |
| Edge-SD-SR (U-Net) | SSIM | **0.5353** | 0.5326 | 0.5265 | 0.5244 | **0.5352** |
| BSR-GAN (CNN-Based) | PSNR | **12.44** | 12.44 | 12.43 | 12.47 | **12.47** |
| BSR-GAN (CNN-Based) | SSIM | **0.3708** | 0.3708 | 0.3704 | 0.3710 | **0.3710** |

**PSNR Results:** Our ACPT approach achieves 19.73 dB, outperforming a baseline with 0% overlap (19.34 dB), representing 99.7% of the upper bound quality (19.78 dB) based on 50% overlap with much lower computational overhead (as shown in the latency breakdown results of Table 4, the impact of ACPT on end-to-end latency is 3.12×). Then, a baseline based on 0% overlap and reflect padding shows substantial degradation of PSNR (18.82 dB), reflecting visible tile boundary glitches, as demonstrated in Figure 8.

**SSIM Results:** ACPT achieves 98.5% of the upper bound performance (0.5389 vs. 0. 5470) based on 50% overlap, demonstrating that our method particularly excels at preserving structural information and perceptual quality.

**Qualitative Comparison of Padding Strategy:** We compare our proposed adjacent padding and the default padding strategy, reflect padding. As shown in Figure 8, reflect padding produces a disjoint between neighbouring tiles, whereas our adjacent padding resolves this issue. This is because adjacent padding provides consistent, contextual information surrounding the tile on which the super-resolution model executes.

In addition, we want to highlight the novelty of ACPT. The concept and goal of using neighbouring tiles or global information shared across some prior works, such as MultiDiffusion (Bar-Tal et al., 2023), DemoFusion (Du et al., 2023) and our work, MobilePicasso. However, MultiDiffusion and DemoFusion use 87.5% and 50% overlaps between tiles to achieve consistency. In contrast, our ACPT aims to reduce overlaps (we achieved 0% overlap) to minimise computational cost while removing glitches with simple yet effective adjacent padding (padding size is merely 6% of the tile), making ACPT a novel solution suitable for efficient and effective mobile deployment of high-resolution image editing.

### F.4 ARCHITECTURE-AGNOSTIC APPLICABILITY OF ACPT

In this subsection, we demonstrate that ACPT is a generalisable technique applicable to both diffusion-based super-resolution models and conventional super-resolution approaches. We demonstrate this by applying ACPT across two substantially different architectures: (1) Edge-SD-SR (Noroozi et al., 2025), a U-Net based diffusion super-resolution model. Note that it is significantly smaller than YONOS-SR (Noroozi et al., 2024) adopted in the upscaling stage of our *MobilePicasso* 3-stage pipeline; (2) BSR-GAN (Zhang et al., 2021), a widely used GAN-based super-resolution model. Note that its architecture is primarily based on convolutional layers, fundamentally different from U-Net based diffusion super-resolution architecture.

As shown in Table 7, our ACPT with adjacent padding consistently achieves over 99.7% of the PSNR performance and 98.5% of the SSIM performance compared to the 50% overlap upper bound baseline across all the evaluated architectures.

### F.5 BASELINE COMPARISON WITH ALTERNATIVE HIGH-RESOLUTION METHODS

In this subsection, we investigated alternative high-resolution image generation methods that rely on the idea of latent projection and late-stage scaling. Among various methods employing the late-stage upscaling approach such as SD-Cascade (Pernias et al., 2024), CDM (Ho et al., 2022), and "Make a

Table 8: Latency and memory usage of the alternative high-resolution baseline for 2K-resolution image generation.

| Device | Method | Latency | Memory |
|--------|--------|---------|--------|
| GPU | Original Text-to-Image | 90.1s | 41.7 GB |
| GPU | Make a Cheap Scaling | 27.9s | **53.0 GB** |

cheap scaling" (Guo et al., 2024), we conducted experiments with "Make a cheap scaling" since it enhances inference computational efficiency.

While this alternative high-resolution baseline achieves some inference speedup, its memory usage increases by 27% reaching 53 GB, indicating this approach is not suitable for mobile devices. For 4K resolution, memory exceeds 80 GB, causing OOM errors even on A100 GPUs with 80 GB VRAM.

## G  EXTENDED RELATED WORK

This paper puts forth a first of its kind diffusion approach via tiling, which allows higher resolution (above $2048 \times 2048$) and privacy preserving low-latency (1–4 secs) on-device deployment for image generation conditioned on text and image inputs. Our work naturally touches upon adjacent fields of text-to-image and (text,image)-to-image applications, under settings of higher and lower resolutions as well as cloud or on-device deployments.

### G.1  TEXT-TO-IMAGE

**Lower resolution cloud deployment**   The holy grail of image generation is to simultaneously achieve fast diverse high-quality sampling. Variational Autoencoders (VAEs), Generative Adversarial Networks (GANs) and Denoising Diffusion Models (DMs) have traditionally only achieved two of the three (Xiao et al., 2021). However, the diverse high-quality samples generated by recent DMs (Song et al., 2021; Rombach et al., 2022; Lipman et al., 2022; Karras et al., 2022; Peebles & Xie, 2022; Lu et al., 2024) captured the imagination of the general public. This sparked a wave of innovation which has decreased DMs sampling time from minutes to seconds across a range of GPU hardware running on desktops or the cloud (Salimans & Ho, 2022; Meng et al., 2022; Liu et al., 2022; Song et al., 2023).

**Lower resolution on-device deployment**   Researchers were quick to notice that many of the latency improvements offered by bleeding edge research work were mostly orthogonal and could be combined to decrease the number of iterations per sample generation as well as to consider smaller and more efficient architectures (Li et al., 2023). Concurrently, GPU-aware optimisations offered additional latency improvements (Chen et al., 2023; Choi et al., 2023). This decreased memory and latency requirements allowing mobile on-device GPU and NPU deployment of lower resolution text-to-image diffusion models, quickly demonstrated with on-device demos (Qualcomm Research, 2023c;a).

Notable recent work considers low bit weights and activations quantisations (e.g., w8a8, w4a8, w4a4) to further reduce on-device requirements (Wang et al., 2024; Li et al., 2025). Furthermore, there is a resurgence of GANs in the context of distillation of pre-trained DMs (Fang et al., 2024), which allow 1-step generation, e.g., SD-Turbo (Stability AI, 2023a) (based on ADD (Sauer et al., 2023)) and MobileDiffusion (Zhao et al., 2023) (based on UFOGen (Xu et al., 2023)). Current state-of-the-art on-device text-to-image generation for $512 \times 512$ is 0.6sec per text prompt (Qualcomm Research, 2023b).

**Higher resolution cloud deployment**   Scaling text-to-image DMs to higher resolution is challenging. High-resolution datasets are limited to LAION5B (Schuhmann et al., 2022), and state-of-the-art models such as Stable Diffusion 2.1 require 200,000 NVIDIA A100 GPU hours with training resolution curricula starting at $256 \times 256$ for hundreds of steps and moving to $512 \times 512$ for thousands of additional optimisation steps. Given the encoder and decoder VAEs robustness to various resolutions as well as the specific convolutional down- and up-scaling architectures of Latent Diffusion Models (LDMs) (Rombach et al., 2022), it has been observed that models trained at $256 \times 256$ or $512 \times 512$

can run inference at higher $512 \times 512$, $512 \times 1024$ and $1024 \times 1024$ resolutions, albeit with noticeable loss of fine grained detail. This has encouraged higher resolution research in adapting pre-trained lower resolution models to higher resolution settings and multiple aspect ratios.

MultiDiffusion (Bar-Tal et al., 2023) demonstrated the effectiveness of leveraging lower resolution pre-trained DMs to enable higher resolution applications. It trains forward and backward overlapped and consistent patch-based denoising models, enabling panoramic high resolution with arbitrary aspect ratios. The generated images have in general high quality with smooth transitions across the overlapping regions. One key limitation is revealed for text prompts requiring the generation of a single object, often resulting in undesired repetition of the object across the generated panorama. ScaleCrafter (He et al., 2024) and DemoFusion (Du et al., 2023) address the aforementioned repetition issues, in a training-free manner. ScaleCrafter (He et al., 2024) achieves higher resolution generation by dilating certain convolutional layer's kernels of the original pre-trained lower resolution model. DemoFusion (Du et al., 2023) does so via an up-sampling, diffusing, and denoising loop with up-scaling skip connections between diffusion processes at different resolutions, leading to higher latency.

Concurrently, fine-tuning to higher resolution was spearheaded by SDXL (Podell et al., 2023), SDXL-Turbo (Stability AI, 2023b), and SDXL-Lightning (Lin et al., 2024) leveraging multi-aspect training with average pixel count around $1024 \times 1024$.

Previous works demonstrated inherent scaling limitations of the first generation of DMs architectures. Initial works curbed latency and memory issues by moving expensive self-attention and cross-attention blocks towards smaller bottleneck layers, as well as replacing expensive gelu operations with cheaper swish alternatives (Choi et al., 2023) and finetuning softmax into element-wise relu (Wortsman et al., 2023). However, significant gains were only demonstrated with more significant changes such as those introduced in HDiT (Crowson et al., 2024) and Würstchen (Pernias et al., 2024). Notably, Würstchen (Pernias et al., 2024) architectural, training and inference improvements decreased the aforementioned 200,000 GPU hours training cost to 24,602 A100 hours.

Linking together the previous works of leveraging pre-trained models, but with adaptor layers for higher resolution at lower training costs Make a Cheap Scaling (Guo et al., 2024) introduces a self-cascade DM with a two stage approach: a pivot guided noise reschedule and a trainable time-aware feature up-sampler. The first approach is reminiscent of DemoFusion (Du et al., 2023) but denoising only at higher Signal to Noise Ratios (SNRs), thereby bypassing the need for skip connections. Training only the time dependent up-sampler layers yields an impressive 5x training speed up with only thousands of parameters. As a trade-off, however, sampling trajectory length and latency increase at inference.

Finally, Li et al. (2024) introduced displaced patch parallelism for leveraging multiple GPUs to decrease the latency of text-to-image generation distributed across multiple GPUs.

### G.2 (TEXT, IMAGE)-TO-IMAGE

**Lower resolution cloud deployment** InstructPix2Pix (Brooks et al., 2022) pioneered the use of image editing based on text prompts. It did so via a two-stage process. Firstly, it generated (text, original image, edited image) tuple datasets via GPT-3 (Brown et al., 2020) and Prompt-to-Prompt (Hertz et al., 2022). Secondly, it modified the architecture of a pre-trained stable diffusion model by doubling the number of channels of the first convolutional layer and feeding the text as well as the stacked original and edited image to the new model for training. Consequently, (text,image)-to-image models memory and computational requirements are higher but nevertheless resemble their text-to-image counterparts, and the body of work mentioned earlier for the latter can often be applied verbatim to the former. This has spurred a range of cloud offerings for lower resolutions synthesis.

**Lower resolution on-device deployment** As an example of the relative cost of image generation based on text and image as opposed to only on text. Current state-of-the-art on-device deployment of (text,image)-to-image (Qualcomm Research, 2024) (via Luo et al. 2023) solutions is around 7sec compared to text-to-image (Qualcomm Research, 2023b) at around 0.6sec.

**High-Resolution Mobile Editing Challenges:** The memory requirements for high-resolution image editing on mobile devices present unique challenges. Tiling-based approaches (Song et al., 2024)

have been explored. While such an approach reduces peak memory, it suffers from significant computational overhead due to tile overlaps. Our work addresses these fundamental limitations through the novel ACPT approach and 3-stage hybrid pipeline, achieving better quality-efficiency trade-offs.

**Higher resolution cloud deployment**    At much higher latency and computational requirements per image generation, Imagic (Kawar et al., 2022) introduced the ability to generate truly complex image edits from demanding text prompts including but not limited to the ability to change the posture and composition of multiple objects in an image. At $1024 \times 1024$ image resolution, their inference process takes up to 8 minutes per image on two TPUv4 accelerators.

## G.3    TEXT-TO-VIDEO

**Lower resolution cloud deployment**    In the context of leveraging pre-trained diffusion models to cater for new applications via adaptors, e.g., for higher resolution (Guo et al., 2024) or additional conditioning (Brooks et al., 2022), VideoLDM (Blattmann et al., 2023) shows that introducing and training temporal and conv3d layers allows higher resolution improvements for text-to-image to translate to higher fidelity video generation.

**Recent mobile video generation:**  On-device Sora (Kim et al., 2025) shows the feasibility of generating short video clips on high-end mobile devices. This work leverages many of the same optimisation strategies developed for image generation while facing additional challenges from temporal consistency requirements and dramatically increased memory demands.

## H    BROADER IMPACT AND FUTURE WORK OF *MobilePicasso*

Our work builds upon these advances while addressing the specific challenge of high-resolution image editing on mobile devices. Unlike previous works focusing primarily on text-to-image generation at standard resolutions, *MobilePicasso* tackles the significantly more challenging problem of 4K image editing within strict mobile constraints.

**Novel Algorithmic Contributions:** Our 3-stage hybrid pipeline can be viewed as a novel form of computational decomposition that makes high-resolution editing tractable on mobile devices. The 3-stage hybrid pipeline and ACPT techniques are complementary to existing acceleration methods and could be combined with recent quantisation advances to achieve even greater mobile efficiency.

**Broader Scientific Impact:** The techniques developed in *MobilePicasso* extend beyond image editing to inform future work in mobile deployment of large-scale AI models. Our approach to cross-domain latent operations, adaptive resource management, and quality-preserving decomposition provides a framework for deploying computationally intensive AI applications on resource-constrained devices.

**Future Work:** In this work, we adopted the pretrained models based on the SDv1.5 architecture because it can be deployed on Snapdragon Hexagon NPU without architectural modifications to reduce the model size. Note that recent architectural advancements such as SD3.5-series (Esser et al., 2024) and Flux.1-series (Black Forest Labs, 2024) models ranging from 8B to 11B parameters exceed the memory capacity of NPUs. Nevertheless, it is worth investigating these advanced pretrained models to further improve the image editing quality.

In addition, further research could combine our approach with recent quantisation approaches (such as INT4, SVDQuant (Li et al., 2025)) and hardware-aware optimisations to achieve even greater mobile efficiency while maintaining the quality advantages demonstrated in our work. Moreover, the impact of this work could extend beyond the immediate application of image editing. We believe that our approach's core techniques (hallucination-aware loss, ACPT, and model-system co-design) are broadly applicable to other generative AI tasks on mobile platforms, including image/sketch-to-image generation, style transfer, and image inpainting/outpainting.

## I    ADDITIONAL IMAGE EDITING RESULTS

In this section, we present additional results that are not included in the main content of the paper due to the page limit.

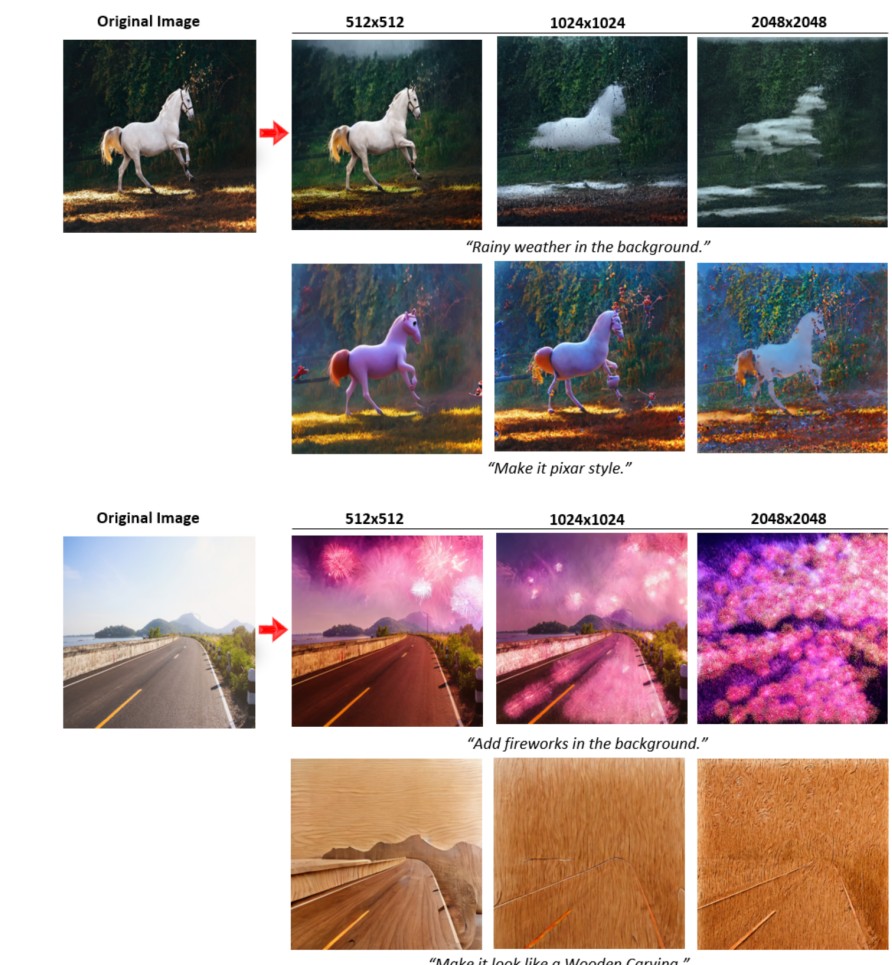

Figure 9: The typical examples of the effect of image resolutions on image-to-image (I2I) generation. As image resolutions get larger starting from $512 \times 512$ to $2048 \times 2048$, I2I image edit models such as IP2P are often unable to produce realistic images that align well with the edit prompt.

## I.1 ADDITIONAL HIGH-RESOLUTION IMAGE EDITING RESULTS BASED ON IP2P

We present additional results of the image editing model, InstructPix2Pix (IP2P) (Brooks et al., 2022), generating edited images for various image resolutions ranging from a standard resolution of $512 \times 512$ to higher resolutions such as $1024 \times 1024$ and $2048 \times 2048$. As image resolutions get larger starting from $512 \times 512$ to $2048 \times 2048$, I2I image edit models often cannot produce realistic images, as demonstrated in Figure 9.

## I.2 ADDITIONAL IMAGE EDITING RESULTS COMPARING *MobilePicasso* AND BASELINES

In this subsection, we provide additional image editing results to demonstrate the effectiveness of our proposed method, *MobilePicasso*, qualitatively. Figures 10 and 11 shows the comparison between our method and all the baselines evaluated in this work.

Input    IP2P    MagicBrush    PIPE    *MobilePicasso*

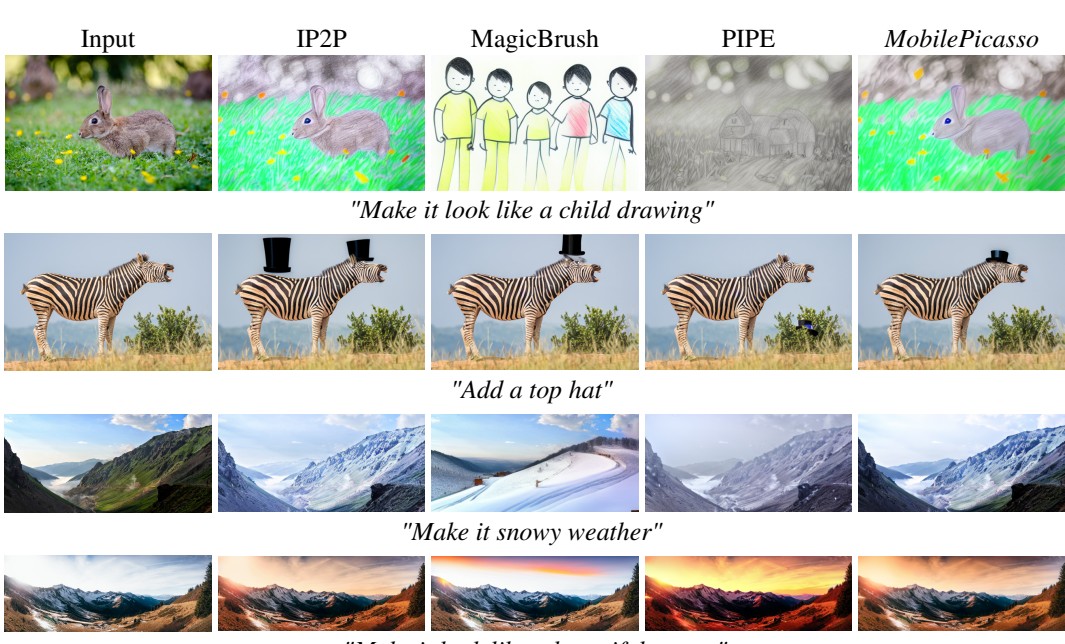

*"Make it look like a child drawing"*

*"Add a top hat"*

*"Make it snowy weather"*

*"Make it look like a beautiful sunset"*

Figure 10: Qualitative comparison among image editing models and *MobilePicasso* given images.

Input    IP2P (w/o Tiling)    IP2P (Tiling, 0% OL)    Ours (ACPT, 0% OL)

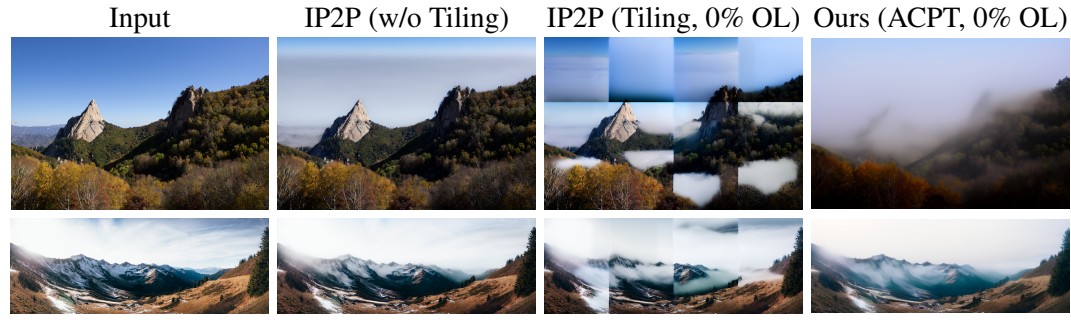

*"Make it covered with fog"*

Figure 11: Qualitative comparison among instruction-based image editing models and *MobilePicasso* for high-resolution input images (2K+). (1) Note that the baselines are based on IP2P without the tiling approach to simulate the server-based image editing which serves as the upper bound. (2) The other baselines are IP2P with 0% tiling ratios. OL indicates an overlap between neighbouring tiles.

