# OpenReview forum: "Efficient High-Resolution Image Editing with Hallucination-Aware Loss and Adaptive Tiling"
_ICLR.cc/2026/Conference — ICLR 2026 Conference Withdrawn Submission_

### Official Review · Reviewer_iD93 · 2025-10-27

**Soundness:** 2
**Presentation:** 3
**Contribution:** 2
**Rating:** 4
**Confidence:** 4

**Summary:**

This paper proposes MobilePicasso, a system for efficient high-resolution (4K) image editing on mobile devices. It introduces a 3-stage hybrid pipeline consisting of (1) standard-resolution image editing with a hallucination-aware loss, (2) learnable latent projection to avoid costly pixel-space operations, and (3) high-resolution upscaling with adaptive context-preserving tiling (ACPT).

MobilePicasso significantly reduces hallucinations and latency while maintaining image quality, achieving up to 55.8× speed-up and only 9% memory increase compared to prior methods. A user study with 46 participants confirms substantial improvements in visual quality and realism over existing diffusion-based image editing models.

**Strengths:**

The paper presents a well-designed and practical solution for high-resolution (4K) on-device image editing, combining a novel 3-stage hybrid pipeline, hallucination-aware loss, and adaptive tiling to greatly improve image quality, reduce hallucinations, and achieve remarkable latency and memory efficiency compared to prior methods.

**Weaknesses:**

1. The novelty of the proposed method appears limited. Although the paper claims that operating in latent space enables efficient high-resolution image editing, similar ideas have already been widely adopted in latent diffusion models. Therefore, the conceptual contribution over existing latent-space pipelines is not entirely clear.


2. The paper omits key engineering aspects such as on-device quantization strategies, exact model compression techniques, or hardware-specific optimizations.

**Questions:**

1. It would be beneficial to include comparisons with training-free image editing methods, such as P2P, Plug-and-Play approaches, or other prompt-based guidance techniques.

2. Does the method works on latest flow models like SD3.5/Flux?

3. In qualitative evaluation, adding examples covering more diverse edit types, such as material and color modification, object removal or relocation, and multi-object semantic edits, would better demonstrate the generality of the proposed method.

---

### Official Review · Reviewer_gSpL · 2025-11-01

**Soundness:** 3
**Presentation:** 3
**Contribution:** 2
**Rating:** 6
**Confidence:** 3

**Summary:**

This paper presents MobilePicasso, a lightweight diffusion-based system for efficient 4K image editing on mobile devices. It improves image quality by up to 48%, reduces hallucinations by 51%, and achieves up to 55× faster performance with minimal memory overhead, even outperforming GPU-based models.

**Strengths:**

The paper effectively identifies the key challenges including hallucination of performing image editing on mobile devices, and the proposed solution appears well-designed and practical. The writing and figures are clear and easy to follow, making the work accessible and well-presented.

**Weaknesses:**

It would be helpful to provide a clearer explanation of the absolute memory and computation requirements needed for real-world deployment on mobile devices. Since your work assumes an on-device setting, the main objective should arguably be to demonstrate practical feasibility rather than primarily comparing performance improvements over prior methods.

**Questions:**

The questions above cover my main concerns, and overall, I find the rest of the paper well-written and technically sound.

---

### Official Review · Reviewer_BLVj · 2025-11-01

**Soundness:** 2
**Presentation:** 2
**Contribution:** 1
**Rating:** 2
**Confidence:** 5

**Summary:**

The paper presents MobilePicasso, a three-stage hybrid pipeline for high-resolution image editing (up to 4K) on mobile devices. On-device high-resolution image editing with diffusion models faces several challenges: (1) the limited resolution of pretrained models, (2) hallucinations in high-resolution image-to-image generation, and (3) limited computational resources on mobile devices. To address these issues, the authors introduce Hallucination-Aware Loss, Data Filtering, and Adaptive Context-Preserving Tiling (ACPT). The three-stage hybrid pipeline consists of: (a) Image Editing Stage, which edits images at low resolution using InstructPix2Pix (with the SD1.5 backbone) and the proposed hallucination-aware loss; (b) Learnable Latent Projection Stage, which trains a lightweight projection model to upscale the latent features, similar to FeatUp; and (c) Upscaling Stage, which generates high-resolution images from the upscaled latents using YONOS-SR with Adaptive Context-Preserving Tiling. The hallucination-aware loss leverages a hallucination detection model (PAL4VST, ICCV 2023) to guide the denoising direction for reducing hallucinations. Adjacent Padding enables 0% tile overlap, while the Adaptive Tiling Overlap Mechanism automatically selects the optimal tiling strategy based on the input resolution. Experiments demonstrate the effectiveness of the proposed framework for high-resolution image editing under resource-constrained settings, both quantitatively and qualitatively.

**Strengths:**

1. The proposed method enables 4K image editing within 42 seconds while utilizing only 1.15 GB of GPU memory.

**Weaknesses:**

1. Limited novelty. Most components appear to be adapted from prior work, and the overall contribution seems largely engineering-oriented rather than conceptually new.
2. I believe Alg. 1 & 2 represent the core contributions of this work. Therefore, they should be moved from the supplementary material to the main paper. Consider relocating the “Formulation of Diffusion-based Image Editing” section (L131–153) to the appendix instead, to make space for these key algorithms in the main text.
3. Identifying defects in the IP2P dataset and improving data quality through preprocessing (filtering by artifact ratio and removing 15% of the training data) is also a small contribution. However, it remains unclear whether the baseline models (especially InstructPix2Pix) in Tab. 1 & 2 were also trained on the same preprocessed dataset. If not, the comparison may not be fair.
4. Running patch-wise editing with InstructPix2Pix and then stitching the results understandably leads to visible seams and implausible images. Wouldn’t running InstructPix2Pix followed by a super-resolution model serve as a more reasonable baseline?
5. Please specify the upper bound of memory capacity typically available on recent mobile devices to contextualize the claimed efficiency.
6. While the proposed method is clearly designed for limited GPU memory, the current output quality is significantly below that of recent SOTA methods. The comparison set also omits several strong baselines. Please include both quantitative (runtime and memory usage) and qualitative comparisons with InfEdit [1] and SwiftEdit [2] to better illustrate the performance gap. If these SOTA models outperform significantly in quality, distillation or quantization could be a more practical direction for deployment.

[1] Xu et al., Inversion-Free Image Editing with Natural Language, CVPR 2024

[2] Nguyen et al., SwiftEdit: Lightning Fast Text-guided Image Editing via One-step Diffusion, CVPR 2025

**Questions:**

Please see the weaknesses.

**Details Of Ethics Concerns:**

No concern.

---

### Official Review · Reviewer_GPpq · 2025-11-01

**Soundness:** 2
**Presentation:** 2
**Contribution:** 2
**Rating:** 6
**Confidence:** 3

**Summary:**

This paper presents MobilePicasso, a system for efficient high-resolution (4K) image editing on mobile devices. It introduces a three-stage hybrid pipeline for standard resolution editing, learnable latent projection, and adaptive tiling. These together with a hallucination-aware loss to reduce visual artefacts. The method achieves significant latency and memory improvements while maintaining high image quality, supported by quantitative results and a user study.

**Strengths:**

1 High-resolution image editing on mobile devices is a practical and important problem.

2 Well-designed three-stage pipeline that balances quality, speed, and memory.

3 Hallucination-aware loss and data filtering effectively improve realism and text alignment.

4 Strong empirical results with large efficiency gains and clear user-study validation.

5 Clear writing and experimental organization.

**Weaknesses:**

1 The definition of hallucinations in this paper is overly simplistic and lacks rigor. In Line 41, hallucinations are described merely as “unrealistic objects or elements generated by diffusion models that were not intended by the edit instruction,” which is too general and subjective. Moreover, the evaluation of hallucinations seems to rely solely on human assessments (“Careful human evaluations show that around 30% of the generated images contain hallucinations”), without a clear or reproducible quantitative metric.

2 The proposed Hallucination-Aware Loss is rather incremental. It simply reuses an existing hallucination detection network (Zhang et al., 2023b) to provide an auxiliary penalty, effectively distilling that model’s capability into the diffusion model. This is a practical but limited extension rather than a novel learning objective or theoretical contribution.

3 The paper mainly focuses on assembling a complete high-resolution image-editing pipeline that can run on mobile devices. While the engineering effort is impressive, the components (hallucination loss, latent projection, adaptive tiling, co-design) are only loosely connected and lack a strong theoretical or algorithmic unification. Each part shows limited novelty on its own, and the overall contribution stems more from system integration than from new learning insights.

**Questions:**

In the user study, approximately how many images did each participant evaluate, and how long did a full session take?
Could such a workload cause potential fatigue effects?
Also, could the authors clarify whether the participants had any background in image processing or computer vision?

---

### Note · Authors · 2025-12-06

I have read and agree with the venue's withdrawal policy on behalf of myself and my co-authors.